# A Data-efficient Neural ODE Framework for Optimal Control of Soft Manipulators

**Mohammadreza Kasaei**
School of Informatics
University of Edinburgh, UK
m.kasaei@ed.ac.uk

**Keyhan Kouhkiloui Babarahmati**
School of Informatics
University of Edinburgh, UK
kkouhkil@ed.ac.uk

**Zhibin Li**
Department of Computer Science
University College London, UK
alex.li@ucl.ac.uk

**Mohsen Khadem**
School of Informatics
University of Edinburgh, UK
mohsen.khadem@ed.ac.uk

**Abstract:** This paper introduces a novel approach for modeling continuous forward kinematic models of soft continuum robots by employing Augmented Neural ODE (ANODE), a cutting-edge family of deep neural network models. To the best of our knowledge, this is the first application of ANODE in modeling soft continuum robots. This formulation introduces auxiliary dimensions, allowing the system's states to evolve in the augmented space which provides a richer set of dynamics that the model can learn, increasing the flexibility and accuracy of the model. Our methodology achieves exceptional sample efficiency, training the continuous forward kinematic model using only 25 scattered data points. Additionally, we design and implement a fully parallel Model Predictive Path Integral (MPPI)-based controller running on a GPU, which efficiently manages a non-convex objective function. Through a set of experiments, we showed that the proposed framework (ANODE+MPPI) significantly outperforms state-of-the-art learning based methods such as FNN and RNN in unseen-before scenarios and marginally outperforms them in seen-before scenarios.

**Keywords:** Soft robots, Non-parametric modelling, Optimal control

## 1 Introduction

Soft robots are composed of compliant materials such as silicone, rubber, or elastomers, which allows them to conform to surfaces and objects while maintaining a level of physical robustness unavailable to their rigid counterparts. This level of compliance makes them suitable for a broad range of applications, including medical procedures, search and rescue operations, and exploration [1]. However, the design and control of soft continuum robots present significant challenges. This is due to the inherent non-linearities and high DOF required to accurately capture the structural deformations that realize these compliant behaviours, which in turn makes it challenging to control the robot's motion. Several methodologies have been proposed to model soft robot controllers, broadly classified into two categories: model-based and data-driven [2].

***Model-based approaches*** rely on mathematical models to represent the dynamics of the robot and use this model to design controllers. Various methodologies have been proposed in this category, including polynomial curvature fitting [3], reduced-order finite element models [4], and lumped parameter models [5, 6]. A comprehensive review of physics-based models for soft robots can be found in [7]. These models have limitations, as they are based on assumptions that may not hold in all conditions and may only be able to accurately describe the behaviour of the robot under a subset of conditions. Furthermore, these methods can be computationally expensive and may not fully capture the nonlinear behaviour exhibited by the robot. Additionally, physics-based models' error tends to increase together with the model inaccuracies, e.g., fabrication inaccuracies.

7th Conference on Robot Learning (CoRL 2023), Atlanta, USA.

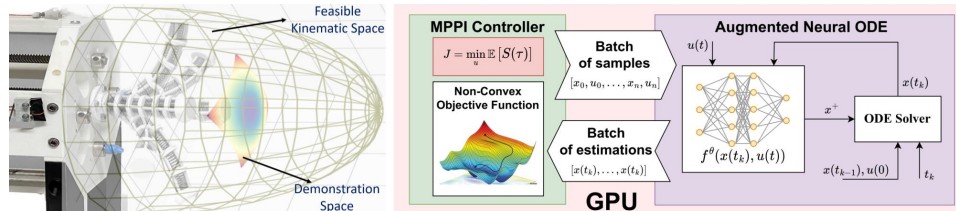

Figure 1: 25 scattered data points are employed within the demonstration space for acquiring knowledge about a continuous forward kinematic model (left). Subsequently, this trained model serves as the foundation for a fully parallel controller capable of managing a non-convex objective function executed on a GPU (right).

***Learning-based approaches*** for soft robot modelling and control utilize collected data to develop models and controllers, either with or without relying on mathematical models. The learning-based models tend to be insensitive to physics-based modelling assumptions and fabrication inaccuracies, especially if training is accomplished on the physical robot arm. These approaches aim to learn kinematic or dynamics models directly [8, 9, 10, 11, 12] or employ model-based/model-free Reinforcement Learning (RL) techniques to learn control policies [13, 14]. While data-driven approaches have the potential to overcome the limitations of model-based methods, they also face inherent drawbacks. First, they require a substantial amount of training data, which can be challenging to acquire for real robots and may compromise their structural integrity as demonstrated in [15], [16] and [17], where 12000, 7000, and 4096 sample points were utilized. Second, the ability to generalize knowledge is limited, leading to poor performance in new or unseen situations [18]. Lastly, data-driven approaches lack interpretability and physical insight, making it challenging to understand decision-making processes and the underlying physical principles of soft robot behaviour [13].

In this paper, we introduce a novel framework for modeling and controlling soft robots. We address the limitations of existing data-driven methods and propose a solution that is extensively tested on a real soft robot in different scenarios. Our key contributions are as follows:

1. We propose a novel approach to developing continuous forward kinematic models for soft continuum robots by employing Augmented Neural ODE (ANODE) [19], a state-of-the-art family of deep neural network models. To the best of our knowledge, this is the first time that ANODE has been applied to modelling soft continuum robots. Our methodology achieves exceptional sample efficiency, training the continuous forward kinematic model using 25 real robot demonstrations.

2. Utilizing the trained model as a basis, we developed a fully parallel Model Predictive Path Integral (MPPI) controller running on a GPU, capable of efficiently managing a non-convex objective function. This controller harnesses the power of parallel processing to optimize trajectory planning and control for the soft continuum robot. By leveraging the capabilities of the trained model, our controller enables the robot to navigate previously unseen trajectories within the feasible kinematic space. In comparison to state-of-the-art data-driven approaches such as feed-forward neural network (FNN) and recurrent neural network (RNN), our framework (FK model and controller combination) demonstrates superior performance in terms of trajectory tracking error while trained on significantly fewer data.

This paper is organized as follows: the methodology of the proposed approach is outlined in Section 2. To evaluate the effectiveness of the proposed framework, a series of experiments are conducted in Section 3, and the results are discussed. An ablation and comparison study with existing approaches will be presented in Section 4. Finally, the conclusion and future work are summarized in Section 5.

## 2 Methodology

This section will provide a comprehensive overview of the key steps involved in our approach. We will start by discussing the process of generating the training dataset, explaining how we collect the necessary data to train our model. Next, we will focus on learning the differential kinematics of the robot, highlighting the techniques and methodologies employed in this process. Finally, we will

describe the development of a controller based on the learned model, explaining how it enables the robot to execute precise and controlled movements.

## 2.1 Training Dataset

The robot utilized in this study is a multi-backbone robot, as depicted in Figure 1 (left). It consists of a central flexible backbone made of a compression spring and four parallel flexible rods encased around it. By manipulating the rods (pulling/pushing), the shape of the robot can be altered. More details about the robot prototype are presented in the appendix. The modeling and controlling of such robots present significant challenges, including the complexity of the coupling between actuation inputs, difficulties in modeling friction between the rods and spacers, particularly when the robot is bent, and the challenge of determining model parameters and the backbone's mechanical characteristics.

In this paper, we intend to utilize a limited dataset to model the behavior of the robot. The dataset was generated by an experienced operator who performed a series of demonstrations using the robot. The operator manipulated the lengths of the rods to move the tip in various directions. The demonstrations were conducted with the rods being pulled/pushed up to 3mm. The feasible kinematic space and demonstration space are depicted in Figure 1. As it is shown, the demonstration space is significantly smaller compared to the feasible kinematic space. The robot inputs, $u_t \in \mathbb{R}^3$, and the corresponding Cartesian coordinates of the robot tip, $x_t \in \mathbb{R}^3$, were recorded at a rate of 15 Hz to generate the training dataset, i.e., $\mathcal{D} = \{x_t^k, u_t^k\}_{k=1}^N$. The position of the robot was estimated using an RGB camera, as discussed in Section 3. It is noteworthy that generating this dataset was efficient and took less than 10 minutes, and our dataset contains N=9100 samples. Only 25 points randomly selected from this dataset will be used to model the robot in the next section, while the remaining data was utilized to train other state-of-the-art machine learning algorithms for comparison.

## 2.2 Learning Differential Kinematics of the Robot

We assume that the robot's behavior can be modeled by a series of nonlinear differential equations:

$$\begin{aligned} \dot{x}(t) &= f(x(t), u(t)), \\ f &: \mathbb{R}^3 \times \mathbb{R}^3 \to \mathbb{R}^3 \end{aligned} \tag{1}$$

with the initial conditions, $x(t_0) = x(0), u(t_0) = u(0)$. It is assumed that a closed-form expression for the function $f$ does not exist. One can use two consecutive states and an action to train a multilayer perceptron (MLP) ($\{x_t, u_t\} \to x_{t+1}$) or a Recurrent neural network (RNN) like nonlinear auto-regressive network with exogenous inputs (NARX). MLPs and NARX models work well on the range of training data but they may struggle with extrapolation, especially if the data lies outside the range of the training data. Additionally, MLPs and NARX models, which typically operate in a discrete-time fashion, may produce predictions that exhibit discontinuities between time steps, leading to less smooth or less physically plausible extrapolations [20].

To overcome these limitations, and to develop a continuous and smooth data-efficient neural network to approximate the robot's model, we formulated the problem as an Augmented Neural ODE (ANODE) [19] which can naturally extend predictions beyond the observed data and time span [20]. Indeed, the robot is modeled using *stiff differential equations* which is characterized by having solutions with rapidly changing components as well as slowly changing components. Explicit methods like Euler (ResNet) [21] will be unstable unless the step size is taken to be extremely small. There are some tricks to overcome these issues but they do not fundamentally change the underlying stability properties. The key features of the ANODE in comparison to the other approaches are data efficiency, capturing complex dynamics, continuous-time formulation, generalization, and handling of trajectory intersections [19]. In this work, we used fixed-adams (Adams-Bashforth-Moulton) [22] method which is an implicit method that is more stable and accurate for stiff differential equations, as they take into account not only the current and previous steps, but also the future step that we are trying to compute. This allows them to handle the rapidly changing components of the solution in a more robust way. Additionally, to prevent trajectories from intersecting, we expand the learning and solution space of the ODE from $\mathbb{R}^3 \times \mathbb{R}^3 \to \mathbb{R}^3$ to $\mathbb{R}^3 \times \mathbb{R}^{3+p} \to \mathbb{R}^{3+p}$. By concatenating a vector of

zeros ($\mathbf{0}_{p \times 1}$) to each data point, we solve the ODE in this augmented space. This augmentation leads to a smoother learned function $f$, resulting in simpler flows that can be computed by the ODE solver in fewer steps. Additionally, it allows the ODE flow to lift points into the extra dimensions, effectively preventing trajectory intersections. This continuous-time formulation allows the model to capture the underlying dynamics and make reasonable extrapolations based on the learned ODEs [19]. With the objective in mind, we proceed to discretize the robot model described in (1) and transform it into a boundary value problem:

$$\mathbf{x}^+ = f_\theta(\mathbf{x}(t), \mathbf{u}(t)), \tag{2}$$

based on the following boundary conditions:

$$\begin{aligned} \mathbf{x}_0 &= \mathbf{x}(t_0), \ \mathbf{u}_0 = \mathbf{u}(t_k), \\ \mathbf{x}_k &= \mathbf{x}(t_k), \ \mathbf{u}_k = \mathbf{u}(t_k), \end{aligned} \tag{3}$$

considering the neural network $f_\theta$ as an approximation of the function $f$, we can determine the solution of $\mathbf{x}(t_k)$ if we have prior knowledge of $f_\theta$:

$$\mathbf{x}(t_k) = \mathbf{x}(t_{k-1}) + \int_{t_{k-1}}^{t_k} f_\theta(x(t), u(t))dt, \tag{4}$$

thus, conventional numerical ODE solvers like the Euler, Runge-Kutta or fixed-adams algorithms can be utilized to estimate the value of $\mathbf{x}(t_k)$:

$$\hat{\mathbf{x}}(t_k) = \text{ODESolver}(f_\theta, \mathbf{x}(t_{k-1}), (t_{k-1}, t_k)). \tag{5}$$

However, in cases where $f_\theta$ is imprecise or remains unknown, it becomes possible to assess the error in estimating the boundary values:

$$\ell = \|\hat{\mathbf{x}}(t_k) - \mathbf{x}(t_k)\|. \tag{6}$$

To update this model, $f_\theta$, we adopt a random selection approach, where only 25 points are chosen from the generated dataset. At each training step, a single point from the dataset is selected. The loss function is estimated using equations (2-6). This error is utilized in a supervised learning manner, and the model is trained using the adjoint sensitivity method [23] to ensure memory efficiency. It is important to note that, based on equation (3), the control inputs remain constant, denoted as $\mathbf{u}_0 = \mathbf{u}_k$, during each training step. To enhance learning efficiency, we have disregarded the dynamics of the control inputs (i.e., $\dot{\mathbf{u}} = 0$) and focused solely on learning the input-output dynamics. Details of training and network architecture are presented in the appendix.

## 2.3 Controller Architecture

In this section, we present a robot control methodology that utilizes the trained model, $f_\theta$, to steer the robot towards arbitrary trajectories in accordance with non-convex cost objectives. The proposed approach is a derivative-free, sampling-based Model Predictive Control (MPC) technique, known as the Model Predictive Path Integral (MPPI) controller, which is capable of handling nonlinear dynamics and non-convex cost objectives [24]. First, a Jacobian matrix, denoted as $\mathbf{J}$, is defined through the utilization of the trained model, $f_\theta$, which maps the velocity of the robot's end-effector to the corresponding velocities in the configuration space:

$$\dot{\mathbf{x}} = \mathbf{J}\dot{\mathbf{u}}, \tag{7}$$

where $\mathbf{J}$ is a $3 \times 3$ matrix, $\mathbf{x} \in \mathbb{R}^3$ and $\mathbf{u} \in \mathbb{R}^3$ represent the robot's end-effector Cartesian coordinates and the input space, respectively. In this work, we employ a numerical estimate of the Jacobian using finite difference method and batch sampling from the trained model. Now, using (7), a discrete-time stochastic dynamical system can be obtained:

$$\mathbf{x}_{t+1} = \mathbf{x}_t + \mathbf{J}(\dot{\mathbf{u}}_t + \delta\dot{\mathbf{u}}_t) \tag{8}$$

where $\delta\dot{\mathbf{u}}_t$ is the control input update which represents by a zero-mean Gaussian noise vector with a variance of $\Sigma_{\mathbf{u}}$ ($\delta\dot{\mathbf{u}}_t \sim \mathcal{N}(\mathbf{0}, \Sigma_{\mathbf{u}})$). Using (8), the control problem can be formulated as a stochastic optimal control problem. Given a finite time-horizon, $t \in \{0, 1, 2, ..., T-1\}$, the objective of the controller is to determine an optimal control sequence, $u = (\mathbf{u}_0, \mathbf{u}_1, ..., \mathbf{u}_{T-1}) \in \mathbb{R}^{m \times T}$, that minimizes

the expected cost-to-go, $S(\tau)$, across all possible system trajectories, $\tau = \{x_0, u_0, x_1, ..., u_{T-1}, x_T\}$, with respect to (8), by taking into account the state-dependent cost function, $S(\tau) \in \mathbb{R}$:

$$J = \min_{u} \mathbb{E}\left[\phi(x_T) + \sum_{t=0}^{T-1}\left(q(x_t) + \frac{1}{2}u_t^T R u_t\right)\right], \tag{9}$$

where $\phi(x_T)$ is the terminal cost, $q(x_t)$ represents running cost and $\mathbf{R} \in \mathbb{R}^{m \times m}$ denotes a positive definite control weight matrix. In order to solve this optimization problem, we adopt an iterative update law, as derived in reference [24], for the implementation of the MPPI algorithm. This algorithm iteratively updates the control sequence for a predefined time horizon, utilizing a successive approximation approach:

$$u_t \leftarrow u_t + \frac{\sum_{k=1}^{K} exp\left(-(1/\lambda)\tilde{S}(\tau_{t,k})\right)\delta u_{t,k}}{\sum_{k=1}^{K} exp\left(-(1/\lambda)\tilde{S}(\tau_{t,k})\right)}, \tag{10}$$

where $K$ represents the number of samples, $\tilde{S}(\tau_{t,k}) = \phi(x_T) + \sum_{t=0}^{T-1}\tilde{q}(x_t, u_t, \delta u_t)$ is a cost-to-go of the $k^{th}$ sample, and $\lambda \in \mathbb{R}^+$ is a hyperparameter called inverse temperature. The cost-to-go function serves as a critical component in guiding the robot's decision-making process, incorporating multiple objectives such as tracking a reference trajectory, obstacle avoidance, and considering affordance. By evaluating the future costs associated with different control inputs, the cost-to-go function allows the robot to anticipate the consequences of its actions. It considers the desired trajectory as a reference, aiming to minimize the deviation from this path while avoiding obstacles. Furthermore, the cost-to-go function can take into account the concept of affordance which encodes the relationships between actions, objects, and their resulting effects [25]. The architecture of the proposed controller is depicted in Figure 1 (right).

# 3 Experiments

Here, a series of experiments were conducted to assess the effectiveness of the proposed approach across various scenarios. Figure 9 shows the experimental setup employed in our study, comprising a flexible and expandable soft manipulator, a Logitech RGB camera positioned within the robot's workspace, and a user interface for recording, initi-

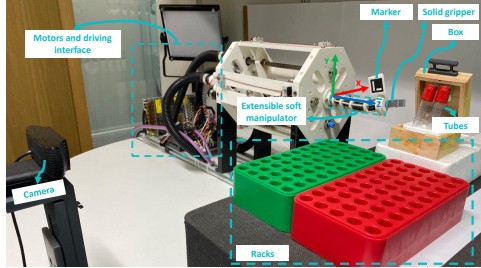

Figure 2: Experiment setup.

ating, and terminating the experiments. To enable the camera to detect the position of the robot's tip, an ArUco marker [26, 27] is affixed to the tip of the robot, providing feedback to the control loop. Details of the robot, the hyperparameters, the controller configuration, and the network structure can be found in the appendix.

## 3.1 Experiment Design

A series of experiments have been conducted to assess the effectiveness of the proposed framework:

- **Static Target Tracking:** ten different targets are pre-defined within the robot's workspace. The objective is to reach these targets with minimal positional error.

- **Trajectory Tracking:** the robot is set to track various trajectories in both 2D and 3D space, which include: **i)** a square on the X-Y plane with sides measuring 0.06 m; **ii)** A circular shape in the XY plane with a radius measuring 0.03 meters; **iii)** A triangle with all sides measuring 0.06 meters; **iv)** An eight-shaped curve defined by the equations $x = a\cos\left(\frac{2t\pi}{T}\right)$ and $y = \frac{b}{2}\sin\left(\frac{4t\pi}{T}\right)$, where $a = 0.03$, $b = 0.05$, $T = 100$ seconds, and $t$ ranges from 0 to 120 seconds; **v)** A helical trajectory is executed along the Z-axis, characterized by a radius of 0.03 m and a pitch of 0.02 m. To assess the tracking's repeatability and accuracy, three trials are conducted for each trajectory.

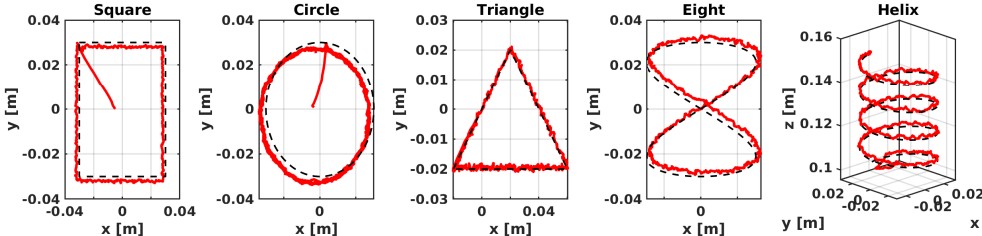

Figure 3: Trajectory tracking results: the robot is set to track diverse trajectories in both two-dimensional (2D) and three-dimensional (3D) space. The solid-red and the dashed-black lines represent the actual and desired trajectories, respectively.

- **Obstacle Avoidance:** in this task, the robot is set to track a helix trajectory characterized by a radius of 0.03 m and a pitch of 0.02 m which is near the border of its feasible kinematic space. The objective of this task is to show how safely the robot is tracking the desired trajectory while avoiding obstacles without being unstable.

- **Box Opening and Test Tube Manipulation:** to showcase a potential application of the robot and the controller's adeptness at incorporating affordance in object manipulation during environmental interactions, we have designed a demanding teleoperation task consists of two sub-tasks: opening the box and picking and placing the test tubes into the rack. In this task, the operator's objective is to maneuver the robot to open the lid of a wooden box, taking into consideration that the robot lacks the strength to counteract the force of gravity and fully extend the lid. This challenging scenario serves as a compelling demonstration of the controller's ability to effectively utilize affordance while navigating the complexities of object manipulation in real-world environments.

### 3.2 Results and Discussions

In the context of static target tracking, the robot's performance is assessed based on three metrics: steady-state error (SSE), standard deviation ($\sigma$), and settling time (ST). The obtained results are showcased in Table 1. It is noteworthy that the position error remains

Table 1: Static Target Tracking Results

|   | SSE (mm) | $\sigma$ (mm) | ST (sec) |
|---|---|---|---|
| $\tilde{x}$ | 0.82 | 0.68 | 5.72 |
| $\tilde{y}$ | 0.71 | 0.58 | 4.21 |
| $\tilde{z}$ | 0.37 | 0.32 | 3.82 |

below the threshold of 1 mm. Additionally, the consistently low standard deviation reveals a stable error pattern observed throughout all experiments.

In trajectory tracking experiments, the evaluation of the robot's performance involves the calculation and presentation of the root mean squared error (RMSE) and standard deviation ($\sigma$) of the error across five distinct trials. These metrics, along with a visual depiction in Figure 3, provide a comparative analysis between the robot's actual tip trajectory and the desired trajectory. Notably, the robot successfully tracks various shapes, including square, circle, triangle, eight, and helix, with a maximum RMSE of $3.2 \times 10^{-3}$ m. Results are summarised in Table 2.

Representative results depicted in Figure 4 showcase the effectiveness of the MPPI algorithm in achieving obstacle avoidance while simultaneously maintaining stable tracking of the desired trajectories. In our implementation, the running cost is the sum of the individual cost terms including a term for tracking a reference trajectory, a term for avoiding obstacles, and a

Table 2: Trajectory Tracking Results

|   | RMSE (mm) | | | $\sigma$ (mm) | | |
|---|---|---|---|---|---|---|
|   | $\tilde{x}$ | $\tilde{y}$ | $\tilde{z}$ | $\tilde{x}$ | $\tilde{y}$ | $\tilde{z}$ |
| Square | 2.40 | 2.60 | 2.50 | 1.40 | 1.50 | 1.30 |
| Circle | 2.80 | 3.20 | 3.10 | 0.51 | 1.50 | 1.50 |
| Triangle | 1.10 | 1.10 | 1.60 | 0.90 | 0.90 | 1.50 |
| Eight | 2.00 | 2.10 | 2.50 | 0.50 | 0.50 | 1.50 |
| Helix | 1.10 | 1.10 | 1.80 | 0.52 | 1.50 | 0.61 |

term for penalizing jerky motions (details are presented in the appendix). The running cost plays a vital role in guiding the robot's behavior by influencing the control actions to minimize deviations and ensure a safe and efficient path. By incorporating the position of the obstacles into the running cost, the robot can effectively evaluate the consequences of its actions and make decisions that prioritize obstacle avoidance and trajectory tracking. This approach enables the robot to dynamically adjust its control inputs based on real-time feedback, resulting in reliable and robust performance even in complex scenarios.

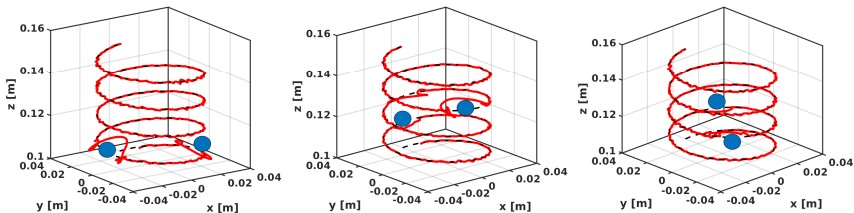

Figure 4: Representative results of obstacle avoidance experiments. Blue dots represent the obstacles, solid-red and dashed-black lines are the actual and reference trajectories, respectively.

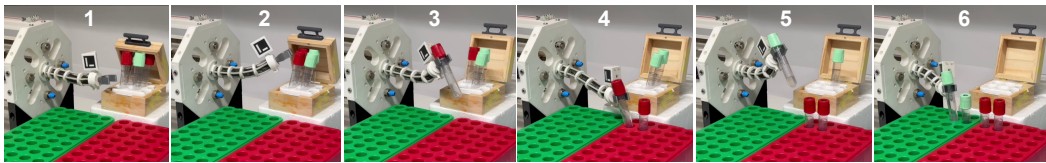

Figure 5: This set of snapshots illustrates the box opening and test tube manipulation task. In this particular task, the operator's objective is to guide the robot in opening the lid of the box and subsequently picking and placing the tubes into the designated racks.

In the Box Opening and Test Tube Manipulation task, the robot operates in a unidirectional tele-operation mode, where the designed controller enables it to track targets specified by an operator through keyboard input. The complexity of this task arises primarily from the absence of direct force feedback, the disparity in kinematics between control interfaces and the robot, the delayed response time, and the lag between the operator's commands and the robot's actions. To address these challenges, we incorporate a set of affordance terms into the running cost of the controllers (details are provided in the appendix). These affordance terms can be selectively activated or deactivated by the operator to limit the motion of the robot along one direction deemed suitable to accomplish the task (i.e, $x$, $y$, or $z$), depending on the task phase. The versatility of MPPI, which can handle non-convex running costs, allows us to effectively utilize these affordance terms for a more intuitive and context-aware interaction between the operator, the robot, and the environment, enabling more effective and efficient teleoperation. A video showing the results is available online https://youtu.be/6tYS-5tkoQg and Figure 5 shows a set of snapshots of this task.

## 4    Ablation and Comparison Study

In this section, we conduct an ablation study to thoroughly compare the performance of the proposed ANODE-based forward kinematics model with Feedforward Neural Network (FNN) and Recurrent Neural Network (RNN) based models in four distinct scenarios. We assess the performance of the models in open-loop and closed-loop trajectory tracking scenarios for both unseen-before and seen-before scenarios, providing a comprehensive analysis of their strengths and limitations. In the open-loop scenarios, we opted to exclude the MPPI controller and instead utilized a simplified approach. Specifically, we employed the equation ($\dot{\mathbf{u}} = \mathbf{J}^+\dot{\mathbf{x}}$), where $\mathbf{J}^+$ represents the pseudo-inverse of the Jacobian and $\dot{\mathbf{x}}$ is the reference trajectory. By implementing this modified strategy, we aimed to observe the system's response without the influence of the MPPI controller, focusing solely on the FK models. In the seen-before scenario, the robot was tasked with tracking an eight-trajectory in X-Y plane within the demonstration space. Conversely, in the unseen-before scenario, the robot was challenged to track a 3D eight-trajectory spanning the entire feasible kinematic space. These distinct scenarios were designed to assess the robot's ability to adapt and generalize its knowledge across different dimensions and spatial configurations. By evaluating its performance in both seen and unseen trajectories, we gain valuable insights into the framework's capacity to handle novel situations and extrapolate its learned behaviors to unfamiliar scenarios.

Furthermore, by comparing ANODE's performance with the traditional MLP and RNN models, we aim to highlight the unique advantages of the ANODE model in capturing the underlying dynamics and temporal dependencies inherent in soft robot forward kinematics prediction. The FNN is constructed with multilayer perceptrons (MLP) to map the input $u_t$ to

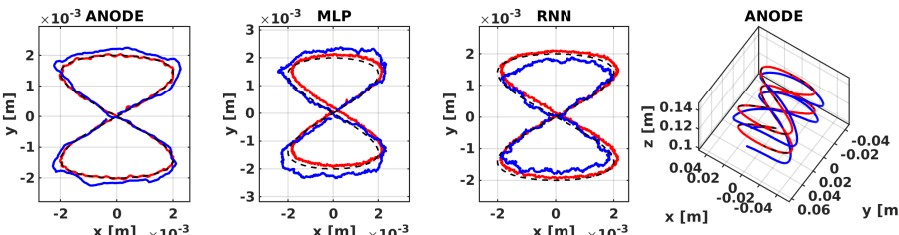

Figure 6: Results of the ablation study: the dashed-black lines, the solid-red and solid-blue lines correspond to the reference, closed-loop and open-loop actual trajectories, respectively.

the output $x_t$. On the other hand, the RNN adopts a NARX architecture, which takes the current state $(x_t, u_t)$ as input and predicts the subsequent state $x_{t+1}$. Both the FNN and RNN models are trained using the dataset $\mathcal{D}$ generated in accordance with the methodology described in Section 2, employing mean square error as their respective loss functions.

To ensure a fair and equitable comparison, we maintained consistent network sizes across all methods. We trained three distinct FK models and integrated them within the identical control loop framework depicted in Figure 1 (right). To evaluate and contrast their respective performances, we computed the RMSE and standard deviation ($\sigma$) of errors across five trials, summarizing the outcomes in Table 3. Representative outcomes are visually depicted in Figure 6,

Table 3: Ablation and comparison results of trajectory tracking scenarios; -O/-C tokens refer to open-loop and closed-loop results, respectively.

| | | RMSE (mm) | | | $\sigma$ (mm) | | |
|---|---|---|---|---|---|---|---|
| | | $\tilde{x}$ | $\tilde{y}$ | $\tilde{z}$ | $\tilde{x}$ | $\tilde{y}$ | $\tilde{z}$ |
| seen | MLP-C | 0.531 | 0.548 | 0.570 | 0.531 | 0.548 | 0.869 |
| | MLP-O | 0.572 | 0.563 | 0.702 | 0.557 | 0.543 | 0.494 |
| | RNN-C | 0.542 | 0.537 | 5.100 | 0.541 | 0.537 | 0.498 |
| | RNN-O | 0.576 | 0.546 | 5.000 | 0.544 | 0.540 | 0.419 |
| | **ANODE-C** | **0.105** | **0.127** | **0.116** | **0.105** | **0.127** | **0.116** |
| | **ANODE-O** | **0.198** | **0.177** | **0.122** | **0.198** | **0.177** | **0.121** |
| unseen | MLP-RNN-(C/O) | - | - | - | - | - | - |
| | **ANODE-C** | **0.256** | **0.164** | **0.157** | **0.256** | **0.164** | **0.157** |
| | **ANODE-O** | **5.600** | **3.100** | **4.900** | **4.700** | **1.800** | **2.900** |

providing further insights into the trajectory-tracking capabilities of each model. As depicted in this figure, all the models demonstrated proficiency in successfully accomplishing the seen-before scenario. Notably, among the tested methods, the ANODE(both open/closed loop versions) showcased superior performance, outperforming the other models in accurately accomplishing the task. In the unseen-before scenario, ANODE stood out as the only model capable of effectively generalizing its knowledge and successfully accomplishing the task. On the other hand, the MLP and RNN models encountered difficulties, leading to the robot becoming uncontrollable. The results showed that the proposed method outperformed alternative approaches significantly in unseen-before scenarios and performed slightly better in seen-before scenarios.

## 5 Conclusion and Limitations

This paper introduced a new method for modelling the continuous forward kinematic models of soft continuum robots using ANODE. The proposed method only required 25 scattered data points. Additionally, we developed a parallel MPPI-based controller running on a GPU, which effectively handles a non-convex objective function. This design enhances the adaptability and robustness of the learned model, enabling accurate prediction and control of soft continuum robot motion in various new scenarios. Through extensive experimentation, ablation, and comparison studies, our proposed framework (ANODE+MPPI) exhibits superior performance over learning-based approaches like FNN and RNN in unseen-before scenarios. It also slightly outperforms them in seen-before settings. While these results are promising, there are some challenges and limitations. The soft robot manipulator being studied has a limited input space comprising three variables. However, when applied to more complex robotic systems characterized by higher dimensions, the Neural ODEs encounters certain limitations. Specifically, the computational costs associated with training Neural ODEs become more challenging. This is attributed to the time-consuming nature of the forward pass, which necessitates the numerical integration of an ODE. Additionally, the proposed methodology assumes the presence of continuous dynamics within the robot, rendering it less suitable for effectively modelling soft robots that exhibit sudden changes, instabilities, or discontinuities in their behaviour.

## 6    Acknowledgement

This work is supported by EU H2020 project Enhancing Healthcare with Assistive Robotic Mobile Manipulation (HARMONY, 101017008) and the Medical Research Council [MR/T023252/1].

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

# Appendix

## 1 Robot Prototype

The robot, as depicted in Figure 9, consists of a flexible backbone rigidly affixed to spacers, accompanied by four rods fixed at the end spacer and passing through the remaining spacers with sufficient clearance, forming the primary body of the robot.

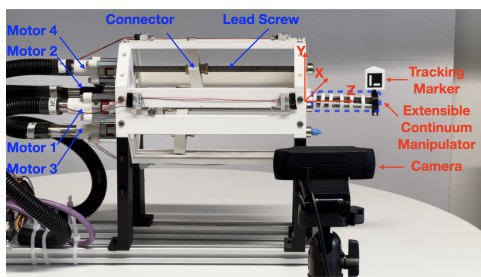

To drive the robot, four brushless DC motors from Maxon Motors, equipped with quadratic encoders and 150:1 reduction gearheads, are utilized. Precise motor position control is achieved through four PID position controller modules (EPOS4 Compact 50/5 CAN), which receive encoder feedback and communicate with a PC using the CAN protocol to establish and retrieve controller set-points and configurations. Lead screws, connected to braided tubes via 3D printed connectors, are attached to the motors to convert motor power into tube-pulling and pushing actions. A schematic of the robot is shown in Figure 8.

Figure 7: Prototype of the flexible robotic arm composed of a reinforced multi-backbone robot. The robot is connected to four brushless DC motors using lead screws. An ArUco marker [26, 27] is placed on the robot tip, and a camera is used to track the marker's position.

## 2 Network Architecture and Training

Table 4 presents a summary of the hyperparameters and network structure. It should be noted that we employed an early-stopping technique to prevent overfitting when training the model. With early stopping, the model's training is halted before it starts to overfit the training data, even if all iterations or epochs have not been completed. This allows the model to avoid memorizing the training data excessively and improves its ability to generalize to new, unseen data.

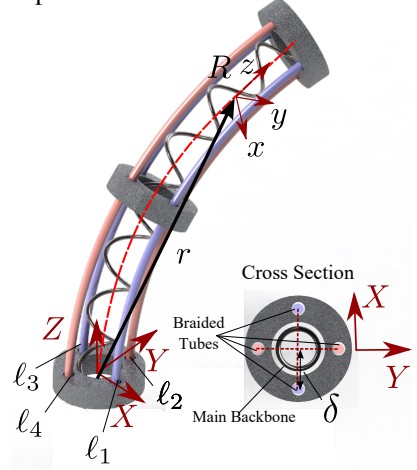

Figure 8: Schematic of the robot.

Table 4: Hyperparameters and network structure.

| Hyperparameter | value |
| --- | --- |
| No. of hidden neuron ($\theta$) | 112 (64,32,16) |
| Augmented vector size (p) | 64 |
| No. of hidden layers | 3 |
| Activation functions | ELU |
| Learning rate | 0.001 |
| Type of ode-solver | fixed-adams |
| Absolute tolerance for ode-solver | 1e-9 |
| Relative tolerance for ode-solver | 1e-7 |
| Number of iteration | 9000 |

## 3 Controller Configuration

This section will provide the details of the controller configurations including its hyperparameters, running cost, and terminal cost functions. The dynamics of the controlled system is captured by the trained FK model (ANODE), while the running cost and terminal state cost are defined as follows:

- **Running cost:** our running cost function is composed of three costs and defined as follows:

$$\text{cost\_tracking} = w_{\text{tracking}} \cdot \|\mathbf{x} - \mathbf{x}_{\text{reference}}\|^2$$
$$\text{cost\_obstacles} = w_{\text{obstacle}} \cdot ((d_1 < 0.01) + (d_2 < 0.01))$$
$$\text{cost\_jerk} = w_{\text{jerk}} \cdot \|\mathbf{u} - \mathbf{u}_{\text{previous}}\|^2$$
$$\text{cost\_affordance} = w_{\text{affordance}} \cdot \text{affordance\_measure}$$
$$\text{running\_cost} = \text{cost\_tracking} + \text{cost\_obstacle} + \text{cost\_jerk} + \text{cost\_affordance}$$

where $\mathbf{x}$ represents the current state of the system, $\mathbf{x}_{\text{reference}}$ is the corresponding state in the reference trajectory, $\mathbf{u}$ denotes the current control input, and $\mathbf{u}_{\text{previous}}$ represents the previous control input. The weights $w_{\text{tracking}}, w_{\text{obstacle}}$, and $w_{\text{jerk}}$ control the importance of each term in the overall cost function. $w_{\text{affordance}}$ determines a suitable metric or measure that quantifies the affordance for the given task or goal. The first term penalizes the deviation of the reference trajectory. These deviations are weighted by a factor of 200, encouraging the system to closely follow the desired trajectory. The second term is a penalty term that considers the distance between the current states and two obstacle locations, denoted as $d_1$ and $d_2$. If the distance to either obstacle is less than 0.01, a high penalty of 100,000 is added. This incentivizes the system to avoid approaching the obstacles too closely. To discourage jerky and abrupt movements, we considered another penalty term. This term penalizes high rates of change in acceleration or control inputs. In our implementation, $w_{\text{jerk}}$ is set to 0.1.

- **Terminal cost:** our terminal cost is defined as: $\text{terminal\_cost} = w_{\text{terminal}} \cdot \|\mathbf{x} - \mathbf{x}_{\text{goal}}\|^2$, where $w_{\text{terminal}}$ is the weighting factor that controls the importance of the terminal cost.

The $\lambda$ parameter was set to 1 to balance the importance between the running cost and terminal state cost. The control inputs were constrained within the range defined by umin = [-0.01,-0.01,-0.01] and umax = [0.01,0.01,0.01]. Gaussian noise with a standard deviation of $\Sigma_{\boldsymbol{u}} = 0.001 * \texttt{torch.eye}(3)$ was added to control samples for exploration. The MPPI optimization process involved generating 500 control samples per iteration, with a prediction horizon of 10 time steps. These parameter values were chosen to achieve effective control performance and can be fine-tuned for specific application requirements.

# 4 Affordance

In the context of robotics, an affordance is a relationship between an actor (i.e., robot), an action performed by the actor, an object on which this action is performed, and the observed effect [25]. The general idea of the affordance theory can be used in robotics to provide some information of mapping between objects, agents and the actions they can take on each other, as there is no unified formalization of it in robotics. In our implementation, we incorporate a set of affordance terms (penalties for violating the motion restrictions) into the running cost of the controllers which can be selectively activated or deactivated by the operator, depending on the task phase. Thanks to the versatility of MPPI, which can handle non-convex running costs, allows us to effectively utilize these affordance terms for a more intuitive and context-aware interaction between the operator, the robot, and the environment, enabling more effective and efficient teleoperation. By adding the affordance measure to the running cost, we give more weight to actions that align with the desired affordance.

# 5 Modeling the Entire Body of the Robot

One can model the entire body of a soft robot as a continuous 3D curve. To this end, the configuration of the main backbone can be defined using a unique set of 3D centroids, $r(s,t) : [0,\ell] \times [0,\infty] \rightarrow \mathbb{R}^3 \times [0,\infty]$, and a family of orthogonal transformations, $\mathbf{R}(s,t) : [0,\ell] \times [0,\infty] \rightarrow so(3) \times [0,\infty]$ where $\ell$ denotes length of the robot. The shape of the main backbone is defined by

$$\boldsymbol{r}'(s,t) = \mathbf{R}(s,t)e_3, \quad \mathbf{R}'(s,t) = \mathbf{R}(s,t)[\mathbf{u}(t)]_\times, \tag{11}$$

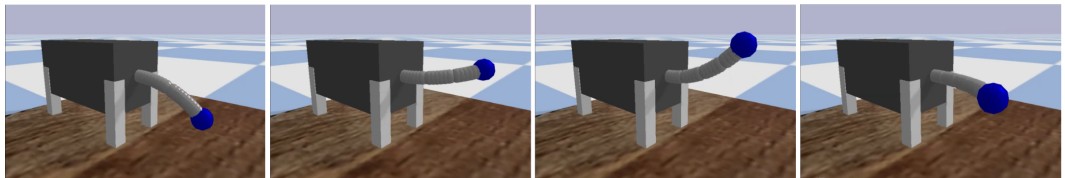

Figure 9: Simulation setup: the simulated robot is tracking a helical trajectory.

$u(t) = [u_x(t), u_y(t), 0]^T$ is the curvature vector of the deformed backbone. $[.]_\times$ operator is the isomorphism between a vector in $\mathbb{R}^3$ and its skew-symmetric cross product matrix, and $e_3 = [0, 0, 1]^T$ is the unit vector aligned with the z-axis of the global coordinate frame. This can be formulated as an ANODE problem and we can calculate the robot end-effector's position as the Cartesian coordinates of the robot's tip:

$$x(t) = r(\ell(t), t) = \int_0^{u_z(t)} r(s, t)\mathrm{d}s. \tag{12}$$

The current system has 14 states including a 3D position, a rotation matrix, and two inputs. To validate the answer, we need a dataset to train a new ANODE, and test its performance. To this end, we developed a simulated version of our robot in the Pybullet simulator and we assumed that the robot can be modeled using Cosserat-rod theory [28]. The code is available online and can be downloaded from here [1]. Figure 9 shows a set of snapshots of the simulated robot while performing a trajectory-tracking task. This simulation enabled the dynamic generation of shape configuration data batches, where we produced 100 random shape configurations by sampling $u_x, u_y$ from uniform distributions $\mathcal{U}(-0.01, 0.01)$ and setting $u_z$ using `torch.linspace(0., 0.05, 100)`. For testing, we expanded the input boundaries to incorporate unseen configurations, adjusting $u_x, u_y$ to sample from $\mathcal{U}(-0.015, 0.015)$ and $u_z$ to span `torch.linspace(0., 0.07, 480)`. For performance assessment, we executed 50 tests, predicting a span of 7 cm, divided into 480 steps. The resulting average Mean Squared Error (MSE) was 0.00001494 which proves the good performance of ANODE to approximate the shape of a soft robot. Two representative results are shown in Figure 10.

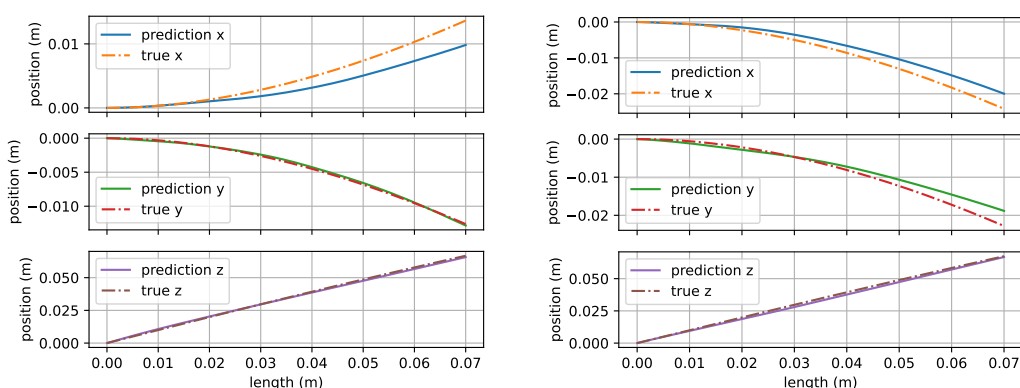

Figure 10: Two representative results of shape reconstruction.

## 6  Neural ODE vs Augmented Neural ODE

Neural ODEs and ANODEs belong to the same family of models. They both use differential equations to model the change in a system over time. The difference between these two lies in how they handle the evolution of the system's state. A Neural ODE allows the state to evolve in the original state space, while an ANODE introduces auxiliary dimensions, allowing the state to evolve in this augmented space. To evaluate their performances in our case, we began by evaluating the NODE's

---

[1]https://github.com/MohammadKasaei/SoftRobotSimulator

capability. We observed a *slight decrease* in performance, particularly in open-loop scenarios with previously unseen conditions. Subsequently, to underscore ANODE's potential in comparison with its counterparts (i.e. MLP, RNN, ResNet), we ventured into a more intricate task — *modeling the robot's entire body* which has been discussed in the previous section. After generating the dataset and training the models, for performance assessment, we executed 50 tests, predicting a span of 7 cm, divided into 480 steps. These predictions were based on control inputs drawn randomly from a uniform distribution $\mathcal{U}(-0.015, 0.015)$ cm, which also covered unseen configurations. The resulting average Mean Squared Error (MSE) for ANODE stood at 0.00001494, distinctly lower than NODE's 0.00172892. A visual comparison of two exemplary outcomes is depicted in Figure 11. Evidently, ANODE outshone in this setup, reaffirming its superior stability and generalization capabilities over NODE.

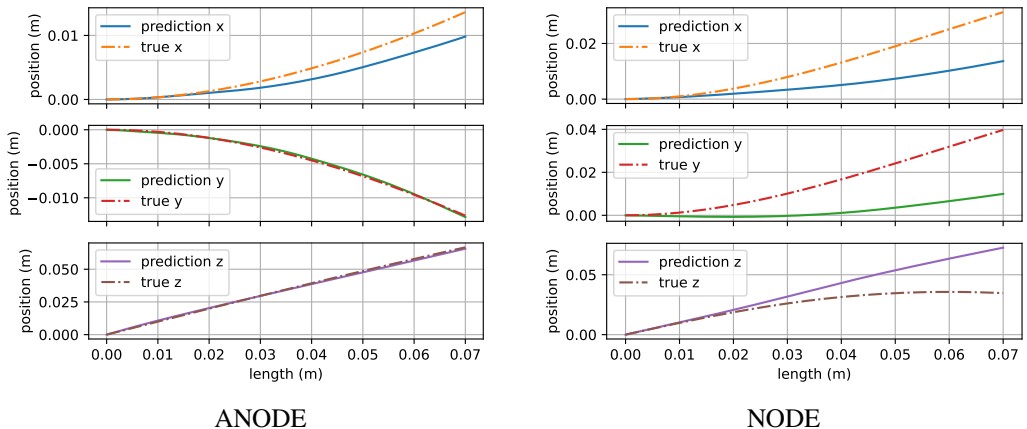

ANODE                                    NODE

Figure 11: Two representative results of shape reconstruction using ANODE and NODE.

