# OpenReview forum: "A Data-efficient Neural ODE Framework for Optimal Control of Soft Manipulators"
_robot-learning.org/CoRL/2023/Conference — CoRL 2023 Poster_

### Official Review · Reviewer_e7Lr · 2023-07-19

**Confidence:** 3
**Originality:** Good
**Technical Quality:** Good
**Clarity Of Presentation:** Good
**Impact:** 3

**Recommendation:**

Weak Accept: I recommend accepting the paper, but will not argue for my recommendation if the majority of other reviewers have a different opinion.

**Review:**

# Strengths
- Differential models have traditionally been used for complex soft robot models, and NODE, which learns differential models in a data-driven manner, seems to have good compatibility.
- The proposed method is applied not only to simple tracking problems but also to practical tasks such as box and test tube manipulation, demonstrating its practicality.
- The comparison between the proposed NODE-based method and other methods like FNN or RNN confirms superior control performance.

# Weaknesses
- In this paper, the authors use Augmented Neural Ordinary Differential Equations (ANODE) for modeling soft robots. However, the reason for using ANODE as a model is not clearly explained. How does modeling soft robots with ANODE address the issues of traditional research?
- NODE introduces many computational expressions that are different from general neural networks due to the handling of differential equations. Can these expressions be effectively utilized in controlling soft robots? If they can be used effectively, clarifying how they are utilized would enhance the value of the paper.

Furthermore, how does the augmentation part of ANODE effectively work in this domain? Additional experiments or supplementary discussions would be beneficial.

**Quality Of The Limitations Section:**

Limitations are addressed clearly

**Questions For Rebuttal:**

In my opinion, the approach of training soft robot models using NODE is beneficial. However, the significance of using NODE for this domain is not clearly explained, and this needs to be clarified.

**Robotics Focus:**

Sufficient demonstration on hardware

**Summary Of Paper:**

This paper proposes a model for efficiently learning a differentiable model in a data-driven manner for controlling soft robots. Specifically, the authors use ANODE to learn a differential model based on a neural network. To generate control command using the model, MPPI is employed. The proposed method demonstrates high control performance in verification experiments using soft robots and shows potential applicability to practical complex tasks.

**Summary Of Recommendation:**

I believe that if the significance of using NODE in the field of soft robotics is clearly explained, it would be a valuable paper worth accepting.

---

### Official Review · Reviewer_pRah · 2023-07-20

**Confidence:** 4
**Originality:** Fair
**Technical Quality:** Good
**Clarity Of Presentation:** Good
**Impact:** 2

**Recommendation:**

Weak Accept: I recommend accepting the paper, but will not argue for my recommendation if the majority of other reviewers have a different opinion.

**Review:**

quality: The quality of the paper could be improved be more details and comparison to other approaches.

clarity: The paper is mostly well written and the presented method is clear

originality: The main idea is to use the ANODE approach as model for the differential kinematics. I don’t think that this approach is original as it just use another type of ML model to learn the kinematics.

significance: As the general pipeline is not new but just with other ML models, the significants of this work might be limited

Strengths:
- The presented approach outperforms the other tested approaches
- The idea is quite simple but effective

Weaknesses:

- A non-learning baseline is missing. For instance, a (simplified) first-principle model could be used.
- It’s not clear why a stochastic MPC approach is necessary
- In general, the paper does not introduce any new methodology but an approach where just another ML model is used.
- Since the model shows good performance for only 25 points, the underlying function must be quite simple. However, there are no insights or discussions on this fact.

**Quality Of The Limitations Section:**

Additional details required

**Questions For Rebuttal:**

- What is the performance of the controller with a Neural ODE instead of ANODE as kinematic model?
- Is there any new methodology?
- Could you please discuss what kind of function f might be?
- Why did you use MPPI?
- Line 94: In which way is this data set "efficient"?

**Robotics Focus:**

Sufficient demonstration on hardware

**Summary Of Paper:**

In this paper, the authors present a data-efficient modeling and control approach for soft manipulators. The main idea is to use augmented Neural ODE (ANODE) model to learn the differential kinematics of the robot. Then, some type of model predicted controller is used to find the optimal input sequence for the actual robot based on the ANODE model. The proposed approach is tested on a soft manipulator and shows superior result in contrast to a plain RNN as feedforward model.

**Summary Of Recommendation:**

The originally of the work is limited as there is no new methodology presented but just another ML model in a standard pipeline.

After rebuttal: Change to "weak accept"

---

### Official Review · Reviewer_p7k6 · 2023-07-21

**Confidence:** 4
**Originality:** Very Good
**Technical Quality:** Very Good
**Clarity Of Presentation:** Very Good
**Impact:** 3

**Recommendation:**

Weak Accept: I recommend accepting the paper, but will not argue for my recommendation if the majority of other reviewers have a different opinion.

**Review:**

Strengths:

- The paper is overall clearly presented and well written.
- Learning-based control of soft robots is a nascent area, and this paper presents a reasonable formulation and empirical results.
- The method is comprehensively benchmarked

Weaknesses:

- Not much insight is provided into why "ANODE" is special for this problem class. What about vanilla ResNets which may be viewed as Euler discretization of Neural ODEs?
- The 25-sample data efficiency is great, but 9K samples were collected - did the performance saturate quickly? That would indicate that the modeling problem is too easy.
- Some baselines for dynamics learning might be missing, e.g., many simulators support deformables - can one use any out of the box?

Other points:

- Figure 1 is way too squished and I never quite understood what the robot looks like and does. Please make it much larger and clearer.
- The dynamics setup is restrictive as the state is simply the end-effector position. What if one wishes to model the entire body of the robot?
- What are physically the control inputs?
- How exactly is the ANODE parameterized?
- Does choice of integrator  (Euler/RK with different stepsizes) make a difference?
- Below Eqn 7, why random selection - with 25 samples, one can do full-batch gradient descent.
- What is capital X-tilde in Table 1? Unless I missed it, I do not see a definition anywhere.
- It is possible that the conclusions of this paper are heavily tied to the robot embodiment with a small state and control space. It is not clear if ANODE will remain preferrerable for high dimensions systems.


**Quality Of The Limitations Section:**

Limitations are addressed clearly

**Questions For Rebuttal:**

Please see questions raised above in the review.

**Robotics Focus:**

Sufficient demonstration on hardware

**Summary Of Paper:**

This paper is a classic application of model-based RL to a soft robotic system. For dynamics learning, the paper uses ANODE (the "dynamics" is really kinematic displacement of the end effector subject to control inputs). For controller, it uses MPPI. The primary conclusion of the paper is the surprising data efficiency of ANODE based dynamics learning relative to MLPs and RNNs.

**Summary Of Recommendation:**

Nice application of ANODEs to soft robot kinematic modeling, and trajectory optimization. The ML parts of the paper are not particularly novel, but the results may be of interest to the soft robot community.

---

### Decision · Program_Chairs · 2023-08-30

**Decision:**

Accept (Poster)

**Comment:**

Thank you to all the reviewers for their careful evaluation and insightful comments. The paper presents a method for applying Augmented Neural Ordinary Differential Equations (ANODE) to soft robot kinematic modeling and trajectory optimization.

All three reviewers agree that the paper is well-written and presented. The paper provides comprehensive benchmarking against other approaches, and its results suggest superior control performance, especially when compared to FNN or RNN-based methods. And that the application of the proposed method to real-world tasks such as box and test tube manipulation demonstrates its practical relevance.

Still, some reviewers noted that while the paper might be technically sound, it does not introduce significant methodological advancements. The primary novelty seems to lie in the application domain rather than the method itself.

Multiple reviewers also suggested that the paper could benefit from including more baselines, especially non-learning ones or those from simulators that support deformables. Although, during the rebuttal, the authors made some arguments, a more concrete quantitative comparison with physics-based simulators will strengthen the paper and increase its relevance.

Considering the aforementioned strengths and concerns, the consensus leans towards a Weak Accept. The application domain, presentation, and empirical results are of value to the community. However, for this work to make a more significant contribution to the community, it would be beneficial for the authors to provide a clearer justification for their methodological choices and possibly explore more diverse baselines.